# Applications of Degradable Hydrogels in Novel Approaches to Disease Treatment and New Modes of Drug Delivery

**DOI:** 10.3390/pharmaceutics15102370

**Published:** 2023-09-22

**Authors:** Bo Hu, Jinyuan Gao, Yu Lu, Yuji Wang

**Affiliations:** 1Department of Medicinal Chemistry, College of Pharmaceutical Sciences, Capital Medical University, Beijing 100069, China; hubo@mail.ccmu.edu.cn (B.H.); jinyuan182@163.com (J.G.); 2Beijing Area Major Laboratory of Peptide and Small Molecular Drugs, Engineering Research Center of Endogenous Prophylactic, Ministry of Education of China, Beijing Laboratory of Biomedical Materials, Beijing 100069, China; 3Laboratory for Clinical Medicine, Capital Medical University, Beijing 100069, China

**Keywords:** degradable, hydrogels, drug delivery, disease treatment

## Abstract

Hydrogels are particularly suitable materials for loading drug delivery agents; their high water content provides a biocompatible environment for most biomolecules, and their cross-linked nature protects the loaded agents from damage. During delivery, the delivered substance usually needs to be released gradually over time, which can be achieved by degradable cross-linked chains. In recent years, biodegradable hydrogels have become a promising technology in new methods of disease treatment and drug delivery methods due to their many advantageous properties. This review briefly discusses the degradation mechanisms of different types of biodegradable hydrogel systems and introduces the specific applications of degradable hydrogels in several new methods of disease treatment and drug delivery methods.

## 1. Introduction

Hydrogels are composed of a three-dimensional polymer network formed by cross-linked chains with a high affinity for water, and these cross-linked chains act as the structural framework of the hydrogels [1]. Furthermore, the resulting pores in the polymer network are able to absorb and retain a large quantity of water, allowing hydrogels to have a very high water content while still retaining their shape. The water content of hydrogels can exceed 95% by weight and can generate small interfacial tension in aqueous environments [2], and such high water content provides physical similarity to body tissue, endowing hydrogels with good histocompatibility. A hydrogel’s polymer network is usually sparse, soft, and flexible while still having desirable mechanical properties.

The first report on the use of synthetic hydrogels for biomedical applications was published in 1960 [3]. Since then, hydrogels have been widely used in many biomedical applications, including drug delivery, tissue engineering, and regenerative medicine, because of their unique properties [4]. Hydrogels are ideal materials for the delivery of biomolecules and even living cells due to their ability to provide the aqueous environment required for biomolecules to function in biological systems. As an attractive drug delivery system, hydrogels have also been used in various branches of medicine, including oncology, immunology, cardiology, and wound healing [5]. However, conventional hydrogels have some limitations in their application. For instance, patients who require subcutaneous implantation of hydrogels for drug delivery require costly surgery and the need to remove the hydrogels after the drugs have taken effect, leading to poor patient adherence [6]. In addition, conductive hydrogels have been widely used in soft, wearable strain sensors due to their excellent electrical conductivity, stretchability, and biocompatibility. However, due to their inherent chemical and physical cross-linking networks, conventional hydrogels are nondegradable, which leads to serious environmental pollution problems [7]. Developing biodegradable conductive hydrogels can help solve the environmental problems caused by such e-waste.

In drug delivery, the loaded drug usually needs to be released slowly over time and must also be able to reach the targeted site. This controlled release in both time and space improves drug targeting and reduces side effects [2]. Fortunately, this can be achieved by degradable crosslinking chains and the type of crosslinking effects inherent in hydrogels [6]. Most conventional hydrogels are cross-linked by covalent interactions, and it is easier to add monomers with multifunctional groups to hydrogels cross-linked by covalent interactions. Nevertheless, once the covalent bond is formed, it is very difficult to break unless the bond forms degradable components. Additionally, matrix degradation is irreversible, which limits the degree to which the degradation characteristics of hydrogels can be controlled. In addition to covalent bonds, other intermolecular interactions provide hydrogels with unique properties.

Many different formulations of degradable hydrogel systems can be created through various interactions such as hydrogen bonding, ionic bonding, and van der Waals interactions, and biodegradable hydrogels are indeed widely used in various areas due to their degradability. For example, researchers [8] have designed a multifunctional epidermal sensor that is stretchable, self-healing, degradable, and biocompatible with nanocomposite-hydrogels that have good potential for human–computer interaction, health diagnostics, and intelligent robotic prosthetics. Moreover, biodegradable hydrogels have an important role in agriculture. The high water absorption of hydrogels allows them to be used to store and retain water and nutrients, particularly in arid areas. They can be used to create slow-release nutrient hydrogel systems, which are important for improving crop yields [9]. Although biodegradable hydrogels have important applications in other fields, in this paper, we are more interested in their applications in the biomedical field, where they are potential candidates for a bevy of biomedical applications due to their biocompatibility and degradability, and their important roles in drug delivery, cell and protein encapsulation, tissue engineering, and wound dressing [10].

With the development of modern medical technology, the treatment of diseases has evolved from single surgical procedures and medications to more effective and safer treatments, though often times more complex treatments. Some examples include photodynamic therapy (PDT), photothermal therapy (PTT), immunotherapy, and gene therapy. PDT is a minimally invasive treatment that causes cell death using photosensitizers that react with oxygen molecules to produce reactive oxygen species in the targeted tissue when excited by certain wavelengths of light [11]. By targeting lesions with precise light, PDT offers improved selectivity compared to conventional treatments. PTT uses the photothermal reaction of a photothermal agent (PTA) to convert absorbed light energy into heat, causing thermal burns in the treated site [12]. PTT is a highly effective and noninvasive treatment involving a simple operation, short treatment time, and short recovery time. Immunotherapy is the tailoring of the host immune response by means of exogenous or endogenous immunity or nonimmune cells and molecules to protect against various diseases [13]. The precise effect of immunotherapeutic agents makes them the most promising therapeutic agents for the treatment of complex diseases. Finally, gene therapy treats diseases by transferring genetic material (DNA or RNA) into the patient’s cells. Genetic material transferred into the cell acts in one of three ways: by expressing the transferred gene, by inhibiting the expression of the targeted gene, or by modifying the targeted gene. Gene therapy can provide lasting cures for diseases through the application of gene transfer or modification [14].

In addition to the development of diverse disease treatment methods, with the rapid advances in drug research and development, conventional forms of drug administration no longer meet the clinical needs of many patients, and several new forms of drug administration have been developed. These include nano-delivery systems, microneedles, targeted delivery systems, and sustained-release delivery systems. Nanodrug Delivery Systems (NDDS) are a class of drug delivery systems composed of nanomaterials that can improve the stability and water solubility of drugs, reduce enzymatic degradation, and increase the rate of uptake by target cells and tissues, thereby improving safety and efficacy [15,16]. In the field of drug delivery, delivering drugs through the skin is called transdermal delivery. Due to the poor permeability of some drugs through the stratum corneum (SC), their application is limited to a few small molecules with high lipophilicity, which usually favors drug molecules with specific physical and chemical properties [17]. To expand the range of molecules that transdermal delivery systems can deliver, microneedles (MNs) have been developed to transiently break or puncture the SC layer by physical or chemical methods. MNs extend the scope of delivery to a wider range of drug molecules and biologics while avoiding the invasive nature of traditional injection methods and improving patient compliance and convenience [17]. Targeted delivery is a new drug delivery system that selectively concentrates drugs in the lesion tissue through a carrier, and this can significantly increase the drug concentration at the lesion site, thus reducing the drug dosage and toxic side effects of a drug to the whole body [18].

In both new approaches to disease treatment and new modes of drug delivery, biodegradable hydrogels play a very important role. A variety of degradable hydrogel systems can be obtained by selecting appropriate hydrogel precursors or complexes according to the desired application. This review summarizes the recent application of biodegradable hydrogels in new disease treatment methods and new modes of drug delivery in order to help researchers and clinicians better understand the potential of biodegradable hydrogels in biomedicine and hopefully provide inspiration for the design of new degradable hydrogels. We briefly introduce the degradation mechanism of biodegradable hydrogels and then analyze the application of biodegradable hydrogels in new disease treatment methods and modes of drug delivery from several aspects. Finally, the application prospects of biodegradable hydrogels in the biomedical field are briefly discussed.

## 2. The Degradation Mechanism of Biodegradable Hydrogels

Biodegradation is the conversion of polymeric materials into simple intermediates or end products by solubilization, simple hydrolysis, or bio-forming entities [19] that results in reduced integrity of the materials and the breakdown of polymers into smaller fragmented molecules. In recent years, biodegradable hydrogels have been widely used in the field of biomedicine. The use of biodegradable systems avoids the hassle of removing the drug delivery system after a drug is fully released. Biodegradable hydrogels can be prepared from synthetic polymers or natural polymers, and since many natural polymers are inherently biodegradable, it is advantageous to use natural polymers to prepare biodegradable hydrogels [20]. Such biodegradable materials need to remain functional throughout their lifetime, so it is necessary to find an appropriate balance between the mechanical and degradation properties of these materials. The degradation process results in the degradation of mechanical properties, and the integrity of the mechanical properties of hydrogels is absolutely necessary for hydrogels to maintain certain unique properties. For example, the swelling properties of hydrogels influence their degradation rate.

To obtain hydrogels with the desired mechanical and degradation properties, the most common approach is to form a two-component system with a hydrophilic swellable component together with a biodegradable hydrophobic component. The former ensures the swelling properties of the hydrogel system, while the latter provides the required degradation and mechanical properties. Crucially, changing the ratio of each component can change the performance of the overall system [19]. Understanding the different degradation mechanisms of hydrogels is thus important for developing biodegradable hydrogel systems. The degradation of hydrogels can be divided into four distinct mechanisms (Figure 1): solubilization, chemical hydrolysis, enzyme-induced degradation, and other mechanisms [19].

(1)Solubilization

Various water-soluble polymers, such as polyethylene oxide and polyvinyl alcohol, can be classified as biodegradable materials based on their water absorption and solubility properties. Here, the degradation process involves the diffusion of water into the hydrophilic polymer to form an expansive system, followed by the dissolution of the polymer in the process of absorbing water [21]. The degree of swelling and dissolution depends on the hydrophilicity of the polymer and the polymer–polymer interaction rather than the polymer–water interaction. Polymer molecular weight, environmental pH, ionic strength, and temperature can all have an effect on solubilization as well.

(2)Chemical Hydrolysis

Any polylactic acid, polyanhydride, polycarbonate, polygenic acid ester, polyhydroxy acid ester, or similar chemicals can be combined with hydrophilic polymers to form hydrogels that exhibit the desired biodegradable properties [22]. Biodegradation of the above polymers occurs via hydrolysis of the ester bonds to form carboxylic acids and alcohols.

(3)Enzyme-Induced Degradation

Enzyme-induced degradation involves a class of hydrolases that catalyze the cleavage of C-O, C-N, and C-C bonds, which is particularly important for the degradation of proteins and polysaccharides in living organisms. Synthetic peptides and poly-alpha-amino acids can also be degraded by enzymes and have been used as both enzymatically degradable drug delivery substrates and biomaterial building blocks. Collagen, gelatin, fibrin gels, and albumin are widely used in the preparation of enzymatically degradable protein hydrogels, and starch and dextran are also widely used in the preparation of biodegradable polysaccharide hydrogels that can be degraded by glycosidases. Some region-specific glycosidases can be used in vivo to develop hydrogels for targeted drug delivery, and here, drugs are released after the hydrogels are degraded by the enzymes in the specific region of delivery. Due to the highly specific binding of enzyme and substrate, enzyme-induced degradation is sensitive to changes that affect enzyme conformation and activity, including pH, ionic strength, and temperature [23]. The diffusion of the enzyme in the hydrogels is also important, and the interaction between the active site of the enzyme and the substrate determines the degradation rate of the hydrogel system. This is particularly important for network systems with a high degree of cross-linking that can cause steric hindrance to enzyme penetration and thereby affect the degradation rate of the system [24].

(4)Other Mechanisms

Some water-insoluble polymers are converted to soluble macromolecules by ion exchange. These polymers mainly include insoluble polyanionic divalent metal salts, such as calcium alginate salt and glycolic acid. When they come into contact with liquids that contain monovalent ions, ion exchange occurs, and water-soluble counterparts are formed [25]. For example, researchers [25] have found that in alginate hydrogels in Na^+^ solution, due to the exchange of Ca^2+^ and Na^+^, the adsorption performance of these composites decreased due to the degradation of the hydrogels, but the hydrogels became stable again in Ca^2+^ solution.

## 3. Applications of Biodegradable Hydrogels in Novel Approaches to Disease Treatment

Due to their good biocompatibility, excellent mechanical properties, and high water content, hydrogels have become practical biomaterials that are widely used in biomedicine (Table 1). Furthermore, Hydrogel systems can be organically combined with other molecules to achieve the desired biological responsiveness [26,27,28]. Some have even said that hydrogels are thusly the first biomaterials to be applied to medicine through deliberate design [29]. Depending on the composition of the polymerization chain and the nature of the cross-linking agent, the properties of a hydrogel can vary greatly, making them versatile biomaterials indeed (Figure 2).

### 3.1. Wound Treatment

#### 3.1.1. Wound Dressing

The skin is the largest, most exposed, and most vulnerable organ in the human body. Once skin damage occurs, the repair process is complex and comprises four classical stages: hemostasis, inflammation, proliferation, and remodeling [30]. Although most skin lesions heal quickly, within one to two weeks, large wounds are often difficult to repair, which can seriously affect health and even threaten people’s lives. Various biomaterials have been developed as wound dressings, including foams, sponges, nanofibers, and hydrogels [31]. Among them, hydrogels have adjustable physicochemical properties due to their similarity to the softness of the extracellular matrix and their ability to fill any irregularly shaped wound. They also act as a hemostatic and sealing agent, preventing fluid leakage from the wound and providing a barrier to bacterial infection, making them ideal wound dressing. However, due to their non-degradability as well as tight adhesion to regenerated skin tissue, removal of the hydrogels can cause secondary damage to the skin and counter the therapeutic effect. Therefore, the possibility of wound dressings based on degradable hydrogels has attracted increasing interest among researchers.

Liang [32] et al. prepared a series of hydrogels for wound dressings using hyaluronic acid-dopamine and reduced graphene oxide (rGO) as a matrix in a hydrogen peroxide/horseradish peroxidase (H_2_O_2_/HRP) system. These hydrogels exhibited high swelling, degradability, tunable rheological properties, and mechanical properties similar to or better than human skin (Figure 3). The antioxidant activity, tissue adhesion, hemostatic ability, and in vivo antibacterial ability of hydrogels enhanced by near-infrared (NIR) radiation of polydopamine have also been investigated, and drug release and inhibition zone experiments demonstrated the sustained drug release ability of the prepared hydrogels. Additionally, hydrogel wound dressings were found to promote angiogenesis, improve granulation tissue thickness and collagen deposition, and stimulate wound healing by upregulating the expression of growth factor CD31.

Qu [33] et al. designed a multifunctional injectable hydrogel dressing to meet the increasing demands of skin injuries. They mixed the biocompatible polymer N-carboxyethyl chitosan (CEC) and oxidized hyaluronic acid–aniline tetramer polymer (OHA-AT) under physiological conditions to produce the conductive antioxidant hydrogel OHA-AT/CEC (Figure 4). This hydrogel has stable rheology, high swelling rate, suitable gelation time, good biodegradability, and free radical scavenging ability. In vivo experiments showed that the OHA-AT/CEC hydrogels significantly accelerated wound healing in a skin defect mouse model and showed good wound repair and skin regeneration ability, providing a new idea for the design of electrically active injectable hydrogels.

#### 3.1.2. Antibacterial Effects

Open wounds and wounds caused by certain skin diseases (such as diabetic foot ulcers) may persist for a long time, and repeated dressing changes and frequent use of disinfectants and antibiotics can lead to the generation of multi-drug-resistant bacteria [32]. A new strategy is, therefore, needed to promote wound healing without cumbersome dressing changes and overuse of antibiotics. Nitric oxide (NO), an endogenous gaseous free radical and neurotransmitter produced by endogenous L-arginine (L-Arg) catalyzed by nitric oxide synthase and its isoenzymes (NOS), mediates wound healing and exhibits broad-spectrum antimicrobial properties [34]. NO regulates cytokines to initiate the inflammatory processes and recruit immune cells to fight microbial infections, does not induce drug resistance, and can promote tissue remodeling [34,35]. With this in mind, Yu [36] et al. established a sustainable wound healing system consisting of a reactive oxygen species (ROS)-responsive hydrogel dressing. The dressing was loaded with the biocompatible NO donor L-Arg (L-Arg@Hydrogel) using hydrogen peroxide (H_2_O_2_) as a trigger. The dressing adsorbs exudate and retains wound moisture upon contacting the wound, and the low level of H_2_O_2_ remaining from wound disinfection reacts with L-Arg in the dressing to trigger the release of NO (Figure 5). Some studies have shown that endogenous levels of L-Arg are insufficient to produce NO under traumatic stress [33]. However, in the L-Arg@Hydrogel system, exogenous L-Arg not only compensated for the lack of endogenous L-Arg in NOS-mediated catalytic NO production but also triggered exogenous NO release in wound tissue. This system not only recruits macrophages to resist microbial infection but also promotes angiogenesis. Furthermore, it induces skin regeneration, thereby promoting wound healing. The L-Arg@Hydrogel system is an ROS-reactive, -biodegradable, and -synergistic wound healing antibacterial hydrogel system with no harmful residues or skin irritation.

#### 3.1.3. Wound Healing

When therapeutic drugs are encapsulated in hydrogels, the hydrogels not only act as a wound dressing but can also enhance the efficacy of the drug. However, designing such hydrogels is complicated, and controlling encapsulation and release is difficult. Chen [37] et al. designed a novel hydrogel that can be used as both a wound dressing and a therapeutic prodrug. They designed a degradable hydrogel based on glutamate and lysine peptides [38] that can be degraded in environments with high levels of degradative enzymes, including tumors and inflammatory sites. The small molecules degraded include glutamate and lysine, two amino acids that have been reported to promote wound healing. Glutamate treatment of wounds has been shown to promote angiogenesis and collagen deposition [38], and lysine deficiency in wounds has been reported to result in failure of collagen deposition [39]. Microneedles (MNs), a minimally invasive and painless transdermal drug delivery system, are widely used in the treatment of a variety of diseases, and MNs that contain active therapeutic drugs have also been used to promote wound healing. The use of MNs increases the contact area with the wound and significantly improves the therapeutic effect of drugs [40]. Yet, MNs made of nonbiodegradable materials can lead to secondary physical damage and microbial infection. Moreover, the active therapeutic components loaded with MNs prepared by biodegradable materials are relatively singular, and some biological agents, such as vascular endothelial growth factor (VEGF) and antimicrobial peptides, are easily inactivated, which limits the application of MNs in the field of wound healing [41,42,43]. One group of researchers [44], therefore, designed a degradable, trauma-friendly hydrogel MN array. They encapsulated zeolite-type imidazolic acid skeleton-8 (ZIF-8) in photocrosslinked hyaluronic acid methacrylate (MeHA) hydrogel, and the encapsulated ZIF-8 was shown to possess potent antimicrobial activity. MeHA is hydrolyzed to low molecular weight hyaluronic acid (HA) in the skin by hyaluronidase, enabling the MN hydrogel array to achieve sustained release of active components at the wound site while avoiding the secondary damage caused by traditional methods. This zinc metal–organic framework (Zn-MOF) encapsulated biodegradable hydrogel MN array can also significantly accelerate epithelial cell regeneration and angiogenesis, providing new ideas for the treatment of wound healing.

Wound infection from various causes, pain, and slow healing of wounds can bring physical and psychological adverse effects to patients. Many methods have been developed to treat different types of wounds, classified according to their form, such as gauze [45,46,47], foam [48,49], and hydrocolloids [50,51], but these methods are still unable to meet certain requirements of maximally effective wound therapy. These methods also frequently result in secondary damage caused by dressing changes, the inability to promote wound healing actively, and inadequate antibacterial properties. However, we still think it is exciting that biodegradable hydrogels could become the best solution to these problems and become a new method for wound treatment!

### 3.2. Tissue Engineering

Tissue engineering, which involves the combination of scaffolding and cell transplantation techniques to design replacement tissues or materials that promote tissue regeneration, is a highly important area of regenerative medicine [52]. Cells, scaffolds, and growth factors are three crucial factors for tissue construction and regeneration, and suitable scaffolds can mimic natural tissues and provide a suitable microenvironment for cell survival, proliferation, and differentiation [53,54]. Hydrogel, a material with high water content and excellent biocompatibility, is widely used as a scaffold structure in tissue engineering [55]. Hydrogels can mimic the structure of the extracellular matrix (ECM) and provide a 3D environment for cell adhesion, ingrowth, and proliferation, facilitating the transfer of soluble nutrients and metabolic waste [56,57]. Conventional hydrogels suffer from low mechanical strength and limited functionality, which has severely limited their application in tissue engineering. Hydrogels based on biodegradable biopolymers such as proteins and polysaccharides have become popular materials for tissue engineering applications due to their excellent biocompatibility, abundant surface functional groups, and ability to immobilize biomolecules. The advent of biodegradable hydrogels has opened up new horizons for tissue engineering scaffolds.

#### 3.2.1. Bone Tissue Engineering

In most situations, bone defects cannot be treated by self-healing, and implants are needed to help repair bone defects. Autogenous bone grafting is the most effective method of bone regeneration due to its good osteoconductivity and osteoinduction [58]. Nevertheless, it has not been widely used in clinical practice because of the possibility of latent infection at the donor site and chronic pain [59]. The clinical application of allogeneic bone transplantation has also been severely hindered by limited donor sources, immune rejection, and potential disease transmission [60]. Bone tissue engineering is a multidisciplinary technology that combines osteoblasts, osteoblast-related factors, and scaffold materials, and it provides a new strategy for the regeneration and repair of bone defects [61]. The process of bone repair is a dynamic physiological process that consists of the synergistic action of various osteogenic factors such as tissue ingrowth, angiogenesis, osteoblast differentiation, and calcium deposition [62]. During the bone repair process, new tissue gradually substitutes bone repair materials. Furthermore, in the process of bone repair, it is necessary to maintain the tissue’s structure and mechanical strength at the repair site until the completion of bone repair and recovery of bone function. Thus, bone repair materials must only gradually degrade [63].

Bone tissue defects caused by fractures and diseases cannot heal spontaneously, so it is necessary to implant biomaterial-based bone substitutes to promote bone tissue regeneration (BTE) [64]. The application of active scaffolds based on hydrogels in BTE has attracted considerable attention. Unfortunately, biopolymer-based hydrogels have poor mechanical strength, relatively rapid degradation, and insufficient osteogenic activity [56], which limits their application in bone regeneration and repair. To regulate the degradation behavior and mechanical properties of biopolymer hydrogels, Zhou [65] et al. designed hybrid gelatin (Gel)/oxidized chondroitin sulfate (OCS) hydrogel using mesoporous bioactive glass nanoparticles (MBGNs) as bioactive fillers for bone regeneration. Adding MBGNs improved the cross-linking of the hydrogels and accelerated the formation of the hydrogels as well. The addition of MBGNs to the hydrogels significantly improved the mechanical properties of the material and increased the storage modulus and compressive strength of the material. MBGN hydrogels also significantly promoted the proliferation and osteogenic differentiation of rat bone marrow mesenchymal stem cells in vitro and the repair of rat cranial defects in vivo (Figure 6). Gel-OSC/MBGN hybrid hydrogels exhibit good mechanical properties and osteogenic activity, which makes them potential injectable biomaterials or scaffold materials for bone regeneration or repair.

Recently, researchers have attempted to add bioactive magnesium ions to bone biological substitutes in order to stimulate the repair of vascularized bone. Only milligrams of these ions can regulate cell behavior, including improving cell adhesion, stimulating cell differentiation, and promoting angiogenesis [66,67] to enhance bone regeneration. Zhang [68] et al. synthesized a polyhedral oligomeric sesquisiloxane group (POSS) double crosslinked hydrogel by the photo-crosslinking of methacryloyl group (GelMA) and coordination of thio-chitosan (TCS) with magnesium ions. This GelMA/TCS/POSS-Mg hydrogels system showed good compatibility between the organic and inorganic components. Zhang et al. also systematically investigated the rheological properties, mechanical properties, morphological structure, and degradation properties of the double cross-linked hydrogel and comprehensively evaluated its cell adhesion, osteogenic differentiation, and angiogenesis ability. They also established a rat skull defect model to further evaluate the osteogenic activity of the composite hydrogel scaffold and demonstrated its ability to promote bone regeneration in vivo.

Bone auto-transplantation is considered to be the best method for the treatment of bone defects. However, the volume of the donor site limits the application of bone auto-transplantation [69]. To improve the healing efficiency of segmental bone defects, Zheng [70] et al. developed a novel multifunctional hydrogel (Figure 7). This hydrogel was prepared by dynamic supramolecular assembly using polyvinyl alcohol (PVA), sodium tetraborate (Na_2_B_4_O_7_), and tetraethyl orthosilicate (TEOS) as raw materials. The reversible network between Na_2_B_4_O_7_ and PVA formed the backbone network of the hydrogel and encapsulated the osteogenesis active ingredients (TEOS). The properties of the supramolecular hydrogels were evaluated by rheological analysis, swelling rate, degradation experiments, and scanning electron microscopy. In in vitro experiments, TEOS hydrogels were self-healing, degradable, showed good biocompatibility, and were able to induce bone marrow mesenchymal stem cells to differentiate into osteoblasts by upregulating the expression of osteopontin (OPN). Additionally, in a rabbit model of segmental bone defect, TEOS-containing hydrogels promoted bone regeneration, restored bone continuity, and recanalized the medullary cavity, indicating the potential of TEOS hydrogel to promote the healing of segmental bone defects.

Phosphate, a major component of bone minerals, plays an important role in bone regeneration [71]. Unfortunately, there has been no systematic focus on calcium- and phosphate-free methods to promote bone regeneration, and phosphorus-containing materials exhibit uncontrolled phosphorus release or significant toxicity, although black Phosphorus (BP) is an allotrope of phosphorus with stable chemical properties and good biocompatibility. Multilayer black phosphorus nanosheets (BPNs) have shown promising applications in biomedicine in recent years, especially in drug delivery, photothermal therapy, and in vivo imaging. Based on the above background, Huang [72] et al. proposed a new strategy to accelerate bone regeneration through a continuous supply of phosphorus without exogenous calcium (Figure 8). They developed a BPNS-based hydrogel system that relied on BPNs to deliver phosphorus steadily and gently. The hydrogel scaffold contained BPNs, and the encapsulated BPNs were able to degrade into phosphorus ions and trap calcium ions to accelerate biomineralization in bone defects. The introduction of BPNs improved the mechanical properties of the hydrogel and accelerated mineralization in vitro, and in vivo results of a rabbit bone defect model showed that BPN hydrogels helped accelerate bone regeneration.

#### 3.2.2. Eye Tissue Engineering

The eyes are one of the most complex organs in the human body and can be divided into anterior and posterior segments. The anterior segment consists of the cornea, conjunctiva, iris, aqueous humor, ciliary body, and lens, and the posterior segment consists of the choroid, sclera, and retinal pigment epithelium [73]. Eye diseases are increasingly causing visual impairment and blindness around the world, and common eye diseases include dry eye, glaucoma, age-related macular degeneration, and retinal diseases [74,75]. Most of these diseases are treated with drugs that have low bioavailability and require frequent administration. The development of biodegradable hydrogels may, therefore, allow a new approach to the treatment of ocular diseases.

Photo-crosslinked hydrogels have excellent biological properties, including in situ injection formation, biocompatibility, biodegradability, and excellent biomechanical properties [76]. The injectable nature of hydrogels makes them minimally invasive and easy to manipulate, and they can also be injected in situ to create the best possible shape match. Currently, the application of hydrogel plug stents is in its infancy, but hydrogel plug scaffolds are expected to gel rapidly after injection and maintain stable mechanical properties.

Dry eye is a common chronic disease, and tear plugs are the most common treatment for dry eye [77]. Tear plugs are inserted in a horizontal position at the opening of the lacrimal punctum or in a deeper position within the lacrimal canaliculus, preventing the drainage of tears and thus retaining tears on the surface of the eyes. However, complications of lacrimal plugs include epiphora, lacrimal plug dysmotility, and infectious canaliculitis, so researchers have attempted to develop more satisfactory lacrimal plugs. Dai [78] et al. reported an injectable silk fibroin hydrogel for application as a degradable tear plug for the treatment of dry eyes (Figure 9). Synthetic indocyanine green fluorescent tracer nanoparticles (FTN) were employed as tracers and methacrylate-modified silk fibroin (SFMA) was used as the hydrogel scaffold. The absorbable tear plugs were formed by rapid photo-crosslinking in situ via free radical polymerization. These SFMA/FTN hydrogel tear plugs possess excellent biocompatibility and biodegradability and can be monitored noninvasively with near-infrared light. Furthermore, in vivo experiments in a rabbit dry eye model showed that SFMA/FTN hydrogel tear plugs could completely block the tear passage and effectively improve the clinical indicators of dry eye.

Retinal pigment epithelial (RPE) cells form the outermost layer of the retina along with Bruch’s membrane [79]. RPE cells play a vital role in the transport of nutrients and metabolic wastes in the retina, and dysfunction or degeneration of RPE cells can contribute to several types of retinal diseases, including retinitis pigmentosa and age-related macular degeneration (AMD). AMD is the major cause of blindness and visual impairment among the elderly in developing countries [79]. Recently, the discovery of the process by which embryonic stem cells (ESCs) differentiate into RPE has stimulated interest in RPE transplantation. Induced pluripotent stem cells (iPSCs) are potential transplant donors that can be generated from adult stem cells without the ethical and legal issues associated with using ESCs. The results of the first clinical trial of autologous iPSC-derived RPE in Japan showed that autologous transplantation could be maintained for at least two years without immunosuppressive therapy [80].

Despite encouraging clinical data, however, the application of adult stem cell-derived RPE in clinical transplantation has been fraught with difficulties. To overcome these difficulties, Gandhi [81] and colleagues utilized human fibrin-derived hydrogels as substrates for RPE transplantation. Here, fibrin cross-linked fiber networks are formed spontaneously after fibrinogen activation. The application of fibrinolytic enzymes allows the degradation of fibrin hydrogel within a few hours, and the rate of degradation can be controlled by changing the fibrinolytic enzymes applied. After cell differentiation, the introduction of fibrinolytic enzymes is able to degrade the fibrin support completely and rapidly, leaving an intact, viable iPSC-RPE monolayer without any adverse effects on cells. Moreover, Han [82] et al. prepared PEG/GG hydrogels with polyethylene glycol (PEG) and gelling glue (GG) to investigate the scaffold’s suitability for cell transplantation. These hydrogels show favorable biocompatibility and degradability and are widely employed in modern tissue engineering. The expression of RPE-specific genes was verified by reverse transcription PCR (RT-PCR), and Han et al.’s results showed that PEG/GG hydrogel had a positive regulatory effect on the expression of RPE-specific genes. Corneal endothelial cells (CECs) are polygonal cells responsible for pumping fluid into the cornea to maintain corneal transparency. Due to their nonregeneration, corneal transplantation is required to restore vision after loss of function [83]. The great demand and limited supply of donor corneas have resulted in a need to treat corneal endothelial dysfunction by culturing and transplanting autologous corneal endothelial cells [84]. Previous studies have shown that the combination of chitosan and PEG can be utilized as a potential scaffold for tissue engineering [85,86], and based on the excellent properties of chitosan and PEG, a team of researchers [84] varying the PEG content, this hydrogel film can achieve the same tensile elongation and ultimate stress as human corneal tissue while maintaining a comparable tensile modulus. What is more, translucency measurements in the visible spectrum showed that the optical transparency of chitosan–PEG hydrogel films (CPHFs) was >95%, which is greater than the transparency of human cornea (maximum 90%). In vitro degradation experiments of bacterial membranes also showed that CPHFs also maintained the biodegradability of chitosan, and in vitro cell culture experiments showed that CPHFs could support the attachment and proliferation of sheep CECs. Finally, in vitro surgical experiments in sheep eyes also showed the excellent performance of CPHFs in implantation, demonstrating the potential of CPHFs in corneal tissue engineering.

#### 3.2.3. Dental Tissue Engineering

Dental treatment approaches have undergone a shift in recent years from the use of synthetic grafts and tissue transplants to tissue engineering through the application of degradable 3D porous hydrogel materials combined with cells or bioactive factors for tissue regeneration, such as dental bone tissue and other oral tissues [84].

Continuous alveolar ridge resorption and soft tissue reduction after tooth extraction not only result in functional and esthetic maxillofacial defects that affect a patient’s quality of life but also compromise subsequent implant placement [87,88]. Currently-used biological scaffolds have poor mechanical properties, fail to degrade rapidly, and are challenging to modify. In contrast, degradable hydrogels possess the biocompatibility, biodegradability, and bone conduction properties required by biological scaffolds for bone regeneration. More importantly, they can also mimic the extracellular matrix of bone and promote cell infiltration and proliferation to form new bone [89,90].

To promote alveolar bone regeneration and reduce alveolar ridge resorption, researchers [91] developed an injectable, degradable polysaccharide hydrogel with a porous structure (Figure 10). This bio-scaffold is an inorganic–organic composite biomaterial scaffold composed of nano-hydroxyapatite and self-healing hydrogel. Polysaccharide-based hydrogels are not only injectable, biocompatible, and biodegradable, but their porous structure allows for the free migration of endogenous stem cells as well, which also allows for irregular bone filling. In a rat alveolar bone defect model, the mandibular central incisor of the rat was extracted, and then scaffold materials were injected minimally invasively at the extraction site. The osteogenic ability of the composite skeleton and the preservation of the alveolar ridge were then evaluated by micro-CT and hematoxylin-eosin (H & E) staining. After four weeks, the new bone area increased by >50%, alveolar ridge preservation was improved by >60%, and traumatic soft tissue healed within one week. This injectable polysaccharide hydrogel composite scaffold is expected to be applied to alveolar bone preservation in the cosmetic field, but it also provides a new strategy for clinical bone defect repair.

Jiang [92] et al. also designed a biodegradable bio-scaffold based on antibacterial chitosan hydrogels for alveolar ridge preservation. They characterized chitosan/polyethylene glycol/polyhexamethylene guanidine hydrochloride (CS/PEG/PHMB) hydrogels with enhanced antibacterial properties by directional porosity, pH sensitivity, and biodegradability. In vitro antibacterial experiments showed that the inhibition rate of staphylococcus aureus was over 90%, effectively inhibiting the spread of bacteria in the alveolar ridge and inflammation. CS/PEG/PHMB hydrogel scaffolds can thus allow the preservation of the alveolar ridge and improve the success rate of dental implants.

Periodontal tissues are the supporting tissues of teeth, in which the periodontal ligament (PDL) plays a role in the structure. Periodontitis induces the loss of periodontal ligament, cementum, and alveolar bone, and periodontal tissue engineering has become a new strategy for reconstructing alveolar bone and for cementum regeneration. Periodontal ligament cells (PDLC) possess stem cell properties and are promising regenerative cells for periodontal tissue engineering [93]. Working from this, researchers [94] have devised a new strategy to deliver PDLC using PEG hydrogels to reconstruct the periodontal ligament: PEG hydrogels as peptide-functionalized EMC for matrix metalloproteinase (MMP)-mediated matrix degradation and PDLC integrin-matrix binding. Their results show that PDLCs in MMP degradable hydrogels expressed core PDL matrix genes and also showed a six- to eight-fold increase in alkaline phosphatase (ALP) activity compared to PDLCs with nondegradable hydrogels as control PDLCs for EMC. The utilization of MMP degradable hydrogels are, thus expected to allow for the application of PDLC in periodontal tissue regeneration.

Hydrogels feature high-water content, 3D polymer networks that are capable of simulating the tissue environment and degrading at specific sites after modification, making them the preferred choice for the therapy of damaged tissues [95]. The applications of biodegradable hydrogels in tissue engineering are not limited to the above examples but are widely employed in various other tissue engineering applications as well.

### 3.3. Cancer Treatment

According to the latest statistics of the International Agency for Research on Cancer (IARC) from the World Health Organization, although the incidence of cancer continues to grow, its mortality rate has continued to decline since 2021 due to the progress in treatment methods [96,97,98]. Currently, there are three conventional cancer treatments: surgery, chemotherapy, and radiation therapy [99], and all three of these methods have limitations. Surgical resection is invasive and carries with it risks of tumor metastasis and infection. Although radiation therapy is a targeted treatment for cancer, it has serious toxic side effects on normal tissues in the vicinity of the irradiated site. Chemotherapy relies on drugs to kill primary and metastatic cancer cells and has become the most common method of cancer treatment [100]. Nevertheless, due to lack of selectivity, most chemotherapeutic drugs generate unsatisfactory treatment outcomes and severe side effects, including liver damage, bone marrow suppression, and immunosuppression [101,102]. With in-depth research of cancer mechanisms and the development of modern medical technology, various new approaches for cancer therapy have emerged, such as immunotherapy, photothermal therapy, photodynamic therapy, gene therapy, and combination therapies thereof.

#### 3.3.1. Biodegradable Hydrogels for Cancer Immunotherapy

Immunotherapy is a treatment that utilizes drugs or modulators to activate and regulate the immune system for disease treatment [103]. Cancer immunotherapy is considered to be the fourth generation of effective cancer treatment after surgery, chemotherapy, and radiotherapy and has broad prospects for improved cancer treatment [104]. Over the past decade, cancer immunotherapy has changed the paradigm of cancer treatment by activating the patient’s own immune system to attack and kill cancer cells. Currently, cancer immunotherapy includes five main categories: immune checkpoint blockade therapy (ICB), lymphocyte-promoting cytokine therapy, chimeric antigen receptor T-cell therapy (CAR-T), activating antibody therapy, and cancer vaccines [105,106,107]. However, routine administration involves high doses or multiple injections, which may introduce safety and efficacy issues [105]. Thus, novel modes of drug delivery are being explored to increase the accumulation of immunotherapeutic agents at the target site to achieve more effective therapy and reduce drug side effects.

Hydrogels play an important role as smart delivery systems for cancer immunotherapy due to their unique physicochemical properties. The high water content of hydrogels provides a suitable physiological environment for “cargo”, their good biocompatibility greatly mimics the properties of natural tissues [108], their soft properties minimize the inflammatory response of surrounding cells [109], and the tunability of their network structures allows them to transport diverse types of substances [110]. Biodegradable hydrogels, in particular, have shown unprecedented potential in cancer immunotherapy [111] (Figure 11).

Glioblastoma (GBM) is one of the tumors with the lowest survival rate of all human cancers [112], and the current treatments for GBM patients are still only radiotherapy and chemotherapy [112]. Most GBM patients inevitably experience tumor recurrence within 6–9 months after initial treatment [113,114], and there is, therefore, an urgent demand for therapies that target GBM tumor recurrence. Immunotherapy is effective in eradicating tumors and triggering immune memory to prevent tumor recurrence, but it has so far failed to improve the prognosis of GBM patients. Hence, researchers [115] established an injectable therapeutic hydrogel degraded by reactive oxygen species for local radioimmunotherapy. Here, the interferon stimulating factor (STING) agonist ADU was employed to induce an immunogenic tumor phenotype, and adeno-associated virus (AAV)-based PD-1 secretion was applied to facilitate subsequent immune responses (Figure 12A,B). In a GBM resection mouse model that mimicked the resection condition of a brain tumor patient, a combination of ADU-AAV-PD1@Gel and radiotherapy was performed (Figure 12C) to test the hydrogel system. The mice generated high levels of reactive oxygen species (ROS) after irradiation, and ROS-degradable hydrogels successfully released therapeutic agents at the implanted tumor sites (Figure 12D). Furthermore, adeno-associated virus serotype 9 was used to express soluble PD-1 (sPD-1) to block the PD-1/PD-L1 pathway. Simultaneously, the released ADU activated the STING pathway and improved tumor immunogenicity and response to therapy. The combination of hydrogel and radiotherapy thus effectively promoted persistent T-cell infiltration, exhibited a potent anti-tumor immune response, and significantly inhibited tumor growth. Furthermore, the combination of ADU-AAV-PD1@Gel with radiotherapy-induced a durable immune memory and prevented the recurrence of GBM. Overall, biodegradable ADU-AAV-PD1@Gel provided a novel strategy for immunotherapy for GBM recurrence after surgical resection.

Sun [116] and colleagues also reported an injectable biodegradable hydrogel containing therapeutic agents for application at the wound site of surgical resection to achieve an in situ immune response. Biodegradable hydrogel scaffolds are capable of rapidly forming hydrogels through a simple hydrogenolysis reaction. These hydrogel scaffolds not only function as sealants to reduce post-operative complications but also serve as vehicles for the delivery of small-molecule therapeutics. Local in situ administration via an injectable hydrogel can downregulate the expression of CD47 on the tumor surface and enhance the function of antigen-presenting cells, stimulating T cell-mediated anti-tumor immunity. This design provides a new strategy for inhibiting tumor recurrence and distant metastasis through the combination of surgery and immunotherapy. Liang [115] et al. prepared a ROS-degradable hydrogel system (GEM-STING@Gel) since gemcitabine (GEM), a traditional first-line chemotherapy drug, could not achieve durable efficacy in the treatment of pancreatic ductal adenocarcinoma (PDAC) alone. This hydrogel system is capable of co-delivering GEM and the STING agonist 5, 6-dimethylflavone-4-acetic acid (DMXAA) to the tumor site. Additionally, it can synergistically activate innate immunity, promote the infiltration of cytotoxic T lymphocytes at the tumor site, and regulate the tumor-suppressive immune microenvironment (TME). In orthotopic surgical models, it has shown a potential to prevent tumor recurrence after surgical resection. This strategy of combining chemotherapy and immunotherapy based on degradable hydrogels not only improves therapeutic performance but also provides superior biological safety.

#### 3.3.2. Biodegradable Hydrogels for Photothermal Cancer Therapy

With conventional cancer treatment, patients can face a high risk of treatment failure and post-treatment side effects [117]. The emerging method of photothermal cancer therapy converts light energy into heat energy through photothermal agents, causing thermal burns at the cancer site, which eliminates various types of cancer and is an effective new treatment [118].

Photo-responsive hydrogels are an ideal platform for controlled-release drug delivery with controlled and less invasive release capabilities. Meng [119] et al. synthesized a nanocomposite of BPNs and hydrogel (BPNs@Hydrogel) to control the release of anticancer drugs under NIR light irradiation (Figure 13). BPN photothermal transducers convert light energy into thermal energy, resulting in an increase in the temperature of the hydrogel matrix. The agarose hydrogel then undergoes reversible softening and hydrolysis, and the anticancer agent can rapidly diffuse from the matrix into the environment, further melting and hydrolyzing under intense near-infrared light and ultimately degrading to oligomers for excretion in the urine. This BPNs@Hydrogel system exhibits excellent therapeutic performance in the treatment of subcutaneous cancer and achieves precise cancer treatment by releasing anticancer agents by external light excitation.

In photothermal therapy, the photothermal conversion agent is the critical component for efficiency. An ideal photothermal conversion agent should possess a high photothermal conversion rate, potent near-infrared light absorption, and excellent biocompatibility [120,121]. Photothermal conversion agents are divided into inorganic photothermal conversion agents and organic photothermal conversion agents. However, many inorganic nanomaterials are challenging to metabolize due to their nonbiodegradability, which results in long-term toxicity that limits their clinical application [122]. Although great progress has been made in the biocompatibility of organic photothermal conversion agents, most of them still present complicated synthesis processes and vague biosafety properties [123]. Consequently, it is desirable to develop photothermal conversion agents that are simple to synthesize and that have excellent biocompatibility in order to achieve broader clinical application of photothermal therapy. To this end, researchers [124] synthesized iodine-starch-alginate (ALG) hydrogel for photothermal therapy via a simple and gentle procedure (Figure 14) based on the classic “iodine-starch test”. ALG is an FDA-approved polysaccharide, and ALG-Ca^2+^ hydrogels are already widely employed in biomedicine [124]. Since all components of iodine-starch-ALG hydrogel are commonly found in clinical practice, the prepared hydrogel exhibited favorable biosafety. In vitro and in vivo experiments demonstrated that the prepared iodine-starch-ALG hydrogel exhibited excellent photothermal conversion ability as well and potent cancer inhibition effects. More importantly, the iodine-starch-ALG hydrogel degraded under human physiological conditions, avoiding potential long-term biotoxicity.

Hydrogels can also be employed to prolong the residence time of photothermal converters in tumors, allowing for multiple photothermal therapies while avoiding the toxicity associated with high doses of such converters [125]. However, this raises a new issue in that the long-term retention of photothermal conversion agents in the body may have adverse effects on healthy tissues. For this reason, researchers [126] designed an injectable and biodegradable nanocomposite hydrogel. The hydrogel was composed of modified glucans and possessed favorable biocompatibility and photothermal effects. Moreover, it was capable of remaining at the tumor site for several days to perform multiple PTTs, eventually achieving complete tumor regression. Subsequently, the hydrogel progressively degraded due to the decomposition of imine bonds, allowing for multiple photothermal treatments and reduced toxicity associated with the long-term retention of photothermal conversion agents.

#### 3.3.3. Biodegradable Hydrogels for Photodynamic Cancer Therapy

Photodynamic therapy (PDT) is a clinically recognized noninvasive cancer treatment modality [127,128,129] that depends on photochemical reactions of photosensitizers under NIR irradiation to generate ROS to damage tumor cells and tissues, especially singlet oxygen [130]. The large amount of reactive oxygen species generated during PDT not only induces apoptosis and necrosis of cancer cells but also allows the ROS-responsive delivery system to release the desired chemotherapeutic agents [131]. The localization of photosensitizers in the human body and the selectivity of photosensitizers at tumor sites are prerequisites for effective tumor killing by PDT. Hydrogels, in turn, are excellent drug reservoirs that can continuously release the loaded drugs or photosensitizers, selectively increasing the drug concentration at the tumor site and reducing the toxic side effects on normal tissues [132,133]. As a result, researchers [134] designed an injectable, photodegradable, and ROS-responsive hydrogel for combined chemotherapy and photodynamic therapy (Figure 15A) by modifying natural hyaluronic acid (HA), which itself features excellent biocompatibility and degradability, and then chemically conjugating the water-insoluble photosensitizer protoporphyrin IX (Pp IX) to the modified HA to form a water-soluble coupling compound. Dynamic acyl hydrazone bonds were formed between the protoporphyrin IX conjugate and the benzaldehyde group of the dialdehyde functional group thioaldehyde ketone (TK-CHO), yielding NIR-triggered ROS degradation hydrogels. As shown in Figure 15B, after intertumoral injection of the precursor solution, hydrogels were formed in situ, and Pp IX selectivity at the tumor site was successfully achieved. The efficacy of PDT was improved, and the side effects on normal tissues were also reduced. Additionally, the ROS generated by PDT achieved the expected degradation of the hydrogel. This injectable NIR-triggered ROS-degradable hydrogel that combines chemotherapy and photodynamic therapy may one day satisfy complex clinical needs. Our brief review of this area of the extant literature shows that although there are numerous reports on the application of hydrogels in photodynamic therapy, there are only a few reports on degradable hydrogels for photodynamic therapy, which is a very promising direction for research.

#### 3.3.4. Biodegradable Hydrogels for Combination Cancer Therapy

Numerous studies have shown that combination therapy is an effective method for the treatment of cancer [135]. As mentioned above, common treatments for cancer include chemotherapy, radiotherapy, immunotherapy, photothermal therapy, photodynamic therapy, and gene therapy, and combination therapy involves the application of two or more of these methods. There are various limitations of monotherapy in the treatment of cancer, and the combination of multiple approaches allows clinicians to compensate for their shortcomings to achieve better treatment outcomes.

An increasing number of studies have shown that the combination of PDT and PTT is capable of exerting synergistic therapeutic effects and minimizing side effects during treatment [136,137,138]. Researchers [139] designed a novel injectable hydrogel based on carbon dots (CDs) and HA for combined PTT and PDT treatment of cancer (Figure 16). Notably, CDs with abundant -NH2 could be employed not only as photosensitizers but also as highly efficient crosslinkers for the formation of hydrogel networks. Biological experiments showed that this hydrogel system exhibited excellent biocompatibility and that the CDs@Hydrogel possessed excellent PDT and PTT effects with a quantum yield of up to 16% and a photothermal efficiency of up to 37%. In vitro and in vivo experiments also showed that it exhibited a satisfactory anti-tumor effect after 660 nm laser irradiation. CDs@Hydrogel was capable of achieving synergistic treatment of PTT and PDT under a single light source, providing new inspiration for more efficient cancer treatment.

Photothermal immunotherapy is an attractive cancer treatment that utilizes a variety of materials to elicit photo-response, and DNA as a delivery biomaterial possesses the inherent properties of biodegradability, biocompatibility, and stimulation of immune response. Wu et al. [140] focused on the development of biodegradable materials for photothermal immunotherapy by designing a DNA CpG hydrogel (DH) loaded with bis-(3′-5′)-cyclic dimeric guanosine monophosphate (G/DH) and coating the formulation with melanin (Mel/G/DH), which was then subjected to elevated temperatures under near-infrared (NIR) irradiation. In vitro, Mel/G/DH + NIR (808 nm) irradiation induced CT26 cancer cells’ exposed surface calreticulin and activated dendritic cells (DC). In vivo, the local application of Mel/G/DH + NIR showed a photothermal killing effect on primary tumors. Mice in the Mel/G/DH + NIR group were five times more likely to survive than those in the Mel/G + NIR group. These findings demonstrate that Mel/G/DH can prevent tumor recurrence by regulating the tumor immune microenvironment. A survey of the literature of this subfield revealed that there are many reports of degradable hydrogels in the combined treatment of chemotherapy and PDT, PTT, and immunotherapy. [115,116,141,142], but the combination of biodegradable hydrogels in PDT, PTT, and immunotherapy is the least frequently reported. Given the encouraging results of Wu et al., for example, we think that it deserves further exploration in the future by a larger number of researchers.

### 3.4. Brain Diseases

Cerebrovascular diseases, including cerebral atherosclerosis, stroke, and Moyamoya disease, are the primary contributors to worldwide all-cause mortality [143,144]. Cardiovascular and cerebrovascular diseases also severely compromise the quality of life of patients and represent an enormous health and economic burden all over the world [145]. To date, clinical treatment options are mainly drugs or other biological agents (such as stem cells or growth factors) [146,147]. However, drugs or biological agents have to overcome physiological barriers to reach the desired site, and therapeutic agents can only play a limited role due to their short residence time at the site of damage. To improve the efficacy and safety of therapeutic agents, it is thus essential to employ advanced drug delivery systems [148], among which degradable hydrogels are one of the most favorable novel drug delivery systems.

Traumatic brain injury (TBI) is a global public health problem that commonly contributes to long-term disability [149,150]. Limited functional recovery after TBI is attributed to progressive neuronal damage following secondary injury-related inflammation [151,152]. Based on their previous work, one group of researchers [153] coupled dexamethasone (DX) with hyaluronic acid to obtain HA-DXM, which reduced neuroinflammation when loaded with degradable hydrogel poly (ethylene glycol-bis-(acryl-oxy-acetate)) (PEG-bis-AA). Furthermore, it improved neuronal survival and function 7 days after injury in a rat model of mild TBI, and an extension of the study showed that the animals treated with PEG-bis-AA/HA-DXM had significantly improved motor function and reduced inflammatory response and lesion volume, making this a promising intervention strategy for TBI therapy.

The study of delivering therapeutic agents to the brain is still an area of ongoing research since it requires spatially and temporally restricted exposure [154]. Several researchers [155] have, therefore, designed a scheme to encapsulate poly(lactic-co-glycolic acid (PLGA) microparticles in degradable PEG hydrogel to achieve the local release of two different types of neurotrophins (NFs) with two separate release curves. The biodegradable hydrogels delivered brain-derived neurotrophic factor (BDNF) and glial-derived neurotrophic factor (GDNF) to Parkinson’s disease-related brain regions, with BDNF localized in the striatum and GDNF localized in the substantia nigra. Compared to the sham-operated group, PEG-degradable hydrogel delivery of NFs significantly reduced microglial response and local inflammation, which is a promising result for reconstructing the substantia nigra striatum.

Moyamoya disease (MMD) is a chronic progressive disease characterized by stenosis or occlusion of the suprascapular carotid artery in either internal carotid artery, accompanied by proliferation of the basilar meningeal collateral [156]. The typical manifestations of MMD disease are transient ischemic attacks (TIAs), ischemic stroke, and intracranial hemorrhage. This disease is not responsive to any medical therapy and can only be cured by surgical revascularization of the cerebral hemispheres through direct or indirect bypass surgery [156]. Nevertheless, patients with MMD typically suffer from a poor prognosis after surgery, and the conditions of arteriogenesis and angiogenesis are unfavorable, which seriously influences recovery [157,158]. Angiogenesis is an essential mechanism of ischemic healing, and improved angiogenesis is thought to contribute to patient recovery directly [159]. Indeed, numerous ischemic animal models have shown that VEGF_165_ can induce angiogenesis and promote revascularization and recovery [160,161,162]. In addition, a number of studies have shown that erythropoietin (EPO) can promote angiogenesis [163,164,165,166], and studies have reported that EPO promotes arteriogenesis better than angiogenesis in the literature [167]. Moreover, researchers have found that the recruitment of monocytes and macrophages to the collateral vessels of arteriogenesis can facilitate arteriogenesis [168,169,170]. Based on this, we propose to design a biodegradable hydrogel to deliver the angiogenic and antiangiogenic growth factors identified above for the prognostic treatment of MMD surgery, a proposition that is currently under investigation.

## 4. Degradable Hydrogels for Novel Modes of Delivery

Traditional hydrogel delivery formulations consist of oral formulations [171] and transdermal formulations [172]. However, oral administration often brings systemic toxicity, and transdermal formulations involve penetrating the skin barrier, which presents various drawbacks of its own. In contrast, biodegradable hydrogels have been developed into diverse novel delivery forms due to their degradable characteristics (Figure 17 [5]).

### 4.1. Injectable Hydrogels

Recently, injectable polymer hydrogels have been employed as delivery vehicles for releasing various anticancer drugs, and they are capable of forming porous three-dimensional networks from free-flowing polymers in response to changes in physiological stimuli (e.g., pH and temperature) [173]. Yet, the large size of microporous hydrogels limits them to releasing only small-molecule anticancer drugs, hampering their application in cancer therapy. Zheng [174] et al. introduced mesoporous silica nanoparticle (MSN) fillers loaded with anticancer drugs to prevent premature release of drugs caused by the wide size of the gaps in the network structure. In addition, MSNs have high specific surface area, suitable size, large pore volume, and stable physicochemical properties. To sustain the release of hydrophobic anticancer drugs, camptothecin (CPT) was encapsulated in the MSNs and subsequently absorbed into physiologically responsive polyethylene glycol-poly (β-amino-ester carbamate) (PAEU) hydrogels, which allowed for longer CPT release times. The MSNs-PAEU copolymer created a stable gel reservoir in the subcutaneous layer of SD rats without toxicity. In addition, the CPT-loaded hydrogels exhibited dose-dependent toxicity to A549 and B16F10 cells. The biocompatibility, biodegradability, and in situ, gel formation properties of the MSN-absorbing PAEU hydrogels make them an ideal drug delivery reservoir for the sustained release of anticancer drugs, and we add that there are quite a few examples of injectable biodegradable hydrogels mentioned above that are not discussed here [33,65,78,90,116,126,141,175,176]

### 4.2. Sprayable Hydrogels

The latest sprayable hydrogels can be pumped through sprayable syringes to form a protective coating that inhibits bacterial growth, providing the dual advantages of portability and in situ rapidity [177,178]. The rapid spraying feature allows closer contact with the surrounding tissues, so it is desirable to select materials with favorable spraying properties and biocompatibility in creating sprays [179]. Sprayable hydrogels have been proposed for cancer therapy and drug delivery platforms in biomedicine due to their various excellent properties. For example, by combining BPNs with thermosensitive hydrogels, Shao [179] et al. developed a sprayable PTT system for postoperative cancer treatment. Their composite hydrogel exhibits photothermal responsiveness to NIR, and its critical gelation time can be precisely controlled by changing the concentration of material components and adjusting the NIR light. Additionally, they established a cancer model to evaluate the potential of the composite hydrogel for cancer treatment after surgery, and the results of animal experiments showed that tumor-bearing mice treated with the composite hydrogel system were completely cured within half a month without tumor recurrence. More importantly, the composite hydrogels possessed excellent biodegradability and biocompatibility, whether in vivo or in vitro. Yuan [180] et al. reviewed the latest advances in sprayable hydrogels and noted their applications in wound healing, postoperative adhesion, and cancer treatment, which we do not repeat here.

### 4.3. Indwelling In Vivo

Biodegradable hydrogels have also been extensively investigated as gastric retention systems for weight loss intervention due to their favorable biocompatibility [181]. Working from this, researchers [182] have developed a biodegradable hydrogel with mechanical stability for long-term gastric retention. Their hydrogel is composed of a double network of polyacrylamide and chitosan/sodium alginate, which is created in the stomach through the swelling of its polyacrylamide network and gradually forms a second network under the action of gastric juice to preserve mechanical stability. Animal models showed that the hydrogel system could remain intact in the stomach for 16 days and subsequently biodegrade in the gastric environment. This prolongs the effect of drugs and reduces the interval between doses, and due to its degradability, it also avoids the problem of poor patient compliance caused by removal. Many of the examples described above overlap with indwelling in vivo and will not be restated [65,70,78,90]. Biodegradable hydrogels have the potential to be developed into a variety of novel delivery forms due to their degradability and biocompatibility, though more novel delivery forms still need to be explored by researchers.

## 5. Conclusions

Hydrogels are a versatile material with multiple applications in biomedicine. Their high water content and biocompatibility allow many possibilities in the area of biological delivery systems, such as the delivery of molecules, cells, and viruses that are not available with other materials. A variety of hydrogel materials can currently be synthesized by free radical polymerization [19], which results in materials that are non- or hardly degradable in vivo, limiting the applications of these materials in many situations. As mentioned in this review, numerous innovative biodegradable hydrogel systems have already been created. These new degradable hydrogel systems possess excellent performance in novel approaches to disease treatment and novel modes of drug delivery.

Biodegradable hydrogels provide an alternative mechanism for releasing therapeutic agents from polymeric matrices, and by regulating the degradation rate of the hydrogels, the release of the loaded therapeutic agents can be controlled. By allowing hydrogels to be degraded only by particular enzymes, it is possible to target hydrogels to release therapeutic agents at specific sites in the human body. However, there is still a broad scope to explore the applications of biodegradable hydrogels in disease therapy and new modes of drug delivery. Furthermore, while degradable sequences have been achieved in hydrogel materials by degradable crosslinkers, the main chains still possess the polymer properties obtained by free radical polymerization [1]. The development of fully biodegradable hydrogel materials is poised to become a key research direction for the future.

## Figures and Tables

**Figure 1 pharmaceutics-15-02370-f001:**
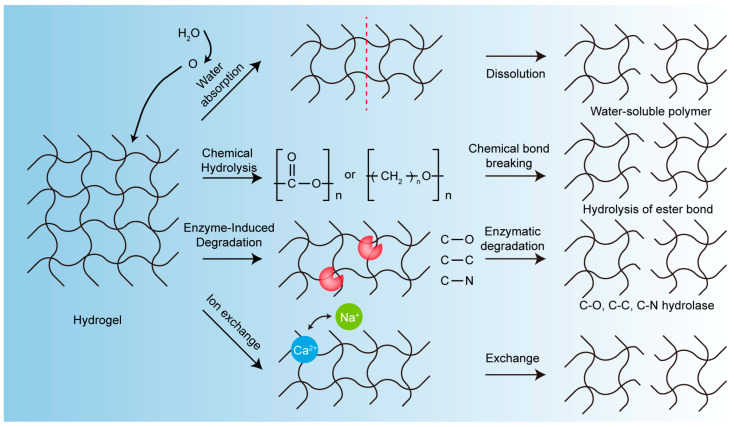
Hydrogel degradation mechanism schematic.

**Figure 2 pharmaceutics-15-02370-f002:**
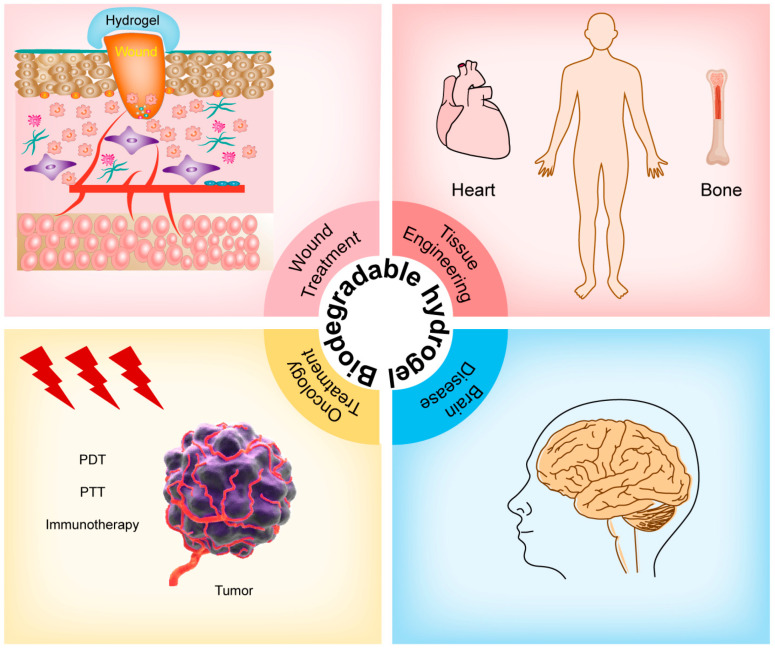
Applications of biodegradable hydrogels in novel approaches to disease treatment.

**Figure 3 pharmaceutics-15-02370-f003:**
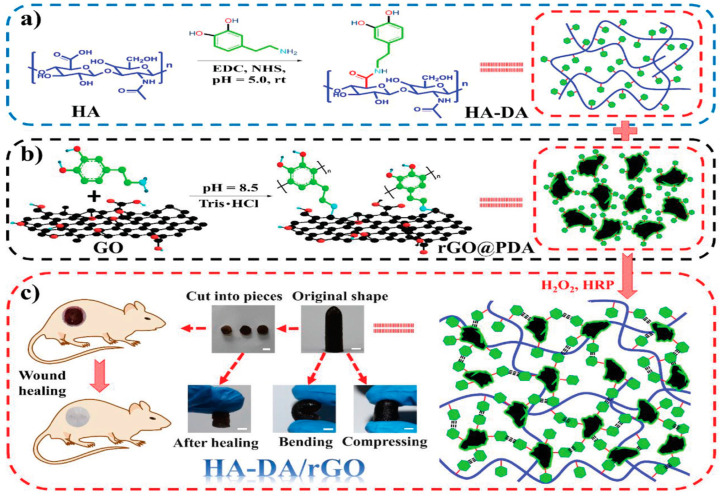
Schematic of HA-DA/rGO hydrogel preparation. (**a**) Preparation scheme of HA-DA polymer; (**b**) rGO@PDA; (**c**) Experimental scheme of HA-DA/rGO hydrogels. HA, hyaluronic acid; EDC, 1-(3-dimethylaminopropyl)-3-ethylcarbodiimide hydrochloride; NHS, N-hydroxysuccinimide; DA, grafting dopamine; HA-DA, hyaluronic acid-graft-dopamine; GO, graphene oxide; PDA, polydopamine; HRP, horseradish peroxidase). Reproduced with permission from [Liang, Y.; Zhao, X.; Hu, T.; Chen, B.; Yin, Z.; Ma, P.X.; Guo, B.], [Small]; published by [WILEY], [2019] [32].

**Figure 4 pharmaceutics-15-02370-f004:**
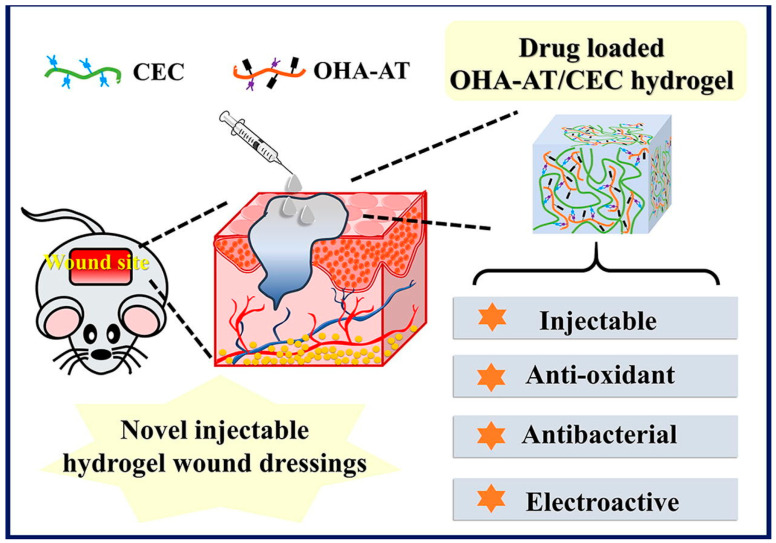
Schematic diagram of OHA-AT/CEC hydrogel experiments. Reproduced with permission from [Qu, J.; Zhao, X.; Liang, Y.; Xu, Y.; Ma, P.X.; Guo, B.], [Chemical Engineering Journal]; published by [ELSEVIER], [2019] [33].

**Figure 5 pharmaceutics-15-02370-f005:**
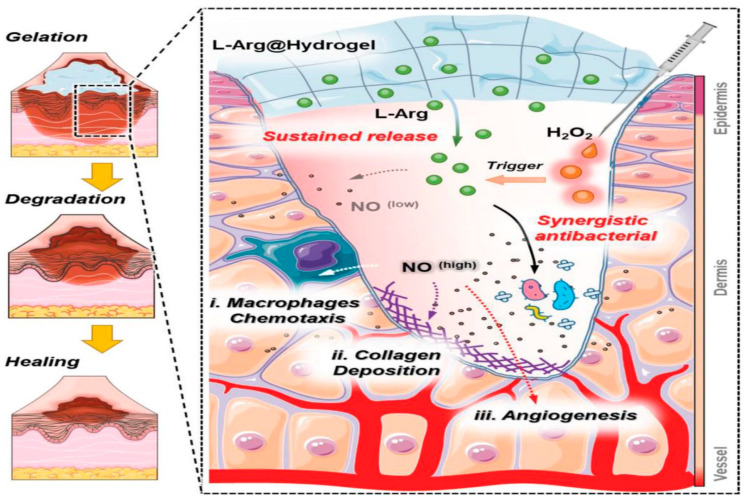
Schematic diagram of the L-Arg@Hydrogel/H_2_O_2_ system for wound healing. Reproduced with permission from [Yu, J.; Zhang, R.; Chen, B.; Liu, X.; Jia, Q.; Wang, X.; Yang, Z.; Ning, P.; Wang, Z.; Yang, Y.], [Advanced Functional Materials]; published by [WILEY], [2022] [36].

**Figure 6 pharmaceutics-15-02370-f006:**
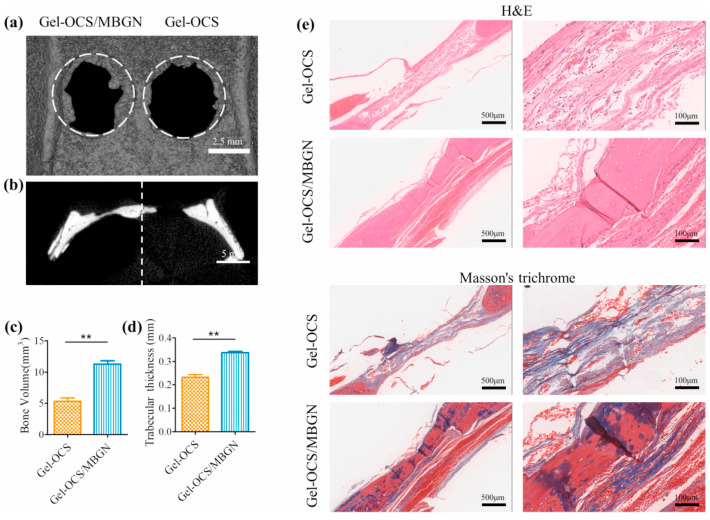
In vivo bone regeneration was observed 6 weeks after Gel-OSC/MBGN hydrogel implantation. (**a**) A typical micro-CT 3D reconstruction image; (**b**) sagittal images of critical rat calvarial defects; (**c**) bone volume (** *p* < 0.01); (**d**) quantitative analysis of trabecular thickness (** *p* < 0.01); (**e**) decalcified skull sections after hydrogel implantation were stained with hematoxylin–eosin staining (H & E) and Masson’s trichrome. Reproduced from [Zhou, L.; Fan, L.; Zhang, F.M.; Jiang, Y.; Cai, M.; Dai, C.; Luo, Y.A.; Tu, L.J.; Zhou, Z.N.; Li, X.J.; et al.], [Bioactive Materials]; published by [KeAi], [2021] [65].

**Figure 7 pharmaceutics-15-02370-f007:**
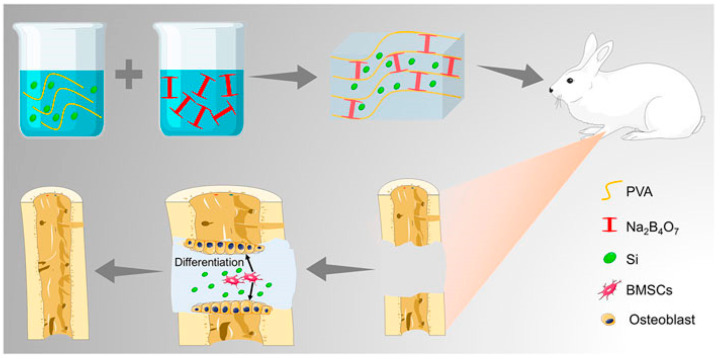
Schematic illustration of how TEOS hydrogels promote the healing of critical large-segment bone defects by inducing osteogenic differentiation of bone marrow mesenchymal stem cells. BMSCs, bone marrow mesenchymal stem cells. Reproduced from [Zheng, S.; Zhong, H.; Cheng, H.; Li, X.; Zeng, G.; Chen, T.; Zou, Y.; Liu, W.; Sun, C.], [Frontiers in Bioengineering and Biotechnology]; published by [Frontiers], [2022] [70].

**Figure 8 pharmaceutics-15-02370-f008:**
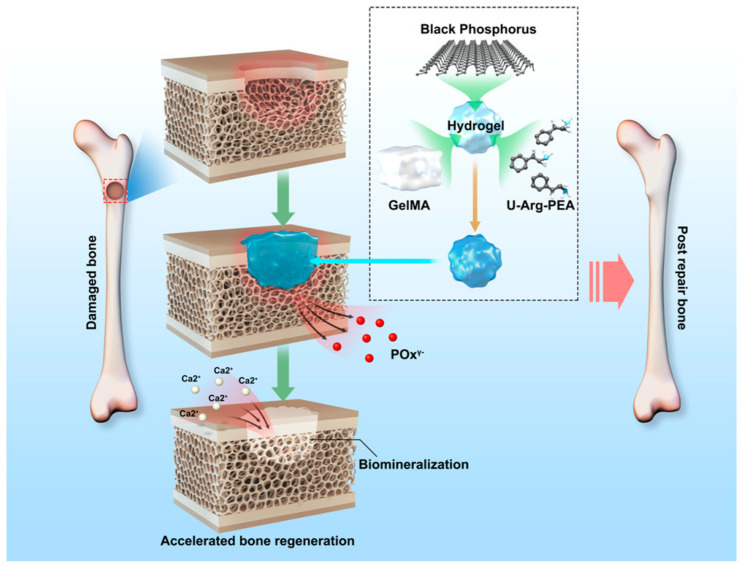
Schematic illustration of BPNs promoting bone regeneration. U-Arg-PEA, cationic arginine-based unsaturated poly(ester amide)s. Reproduced with permission from [Huang, K.; Wu, J.; Gu, Z.], [ACS Applied Materials & Interfaces]; published by [ACS Publications], [2019] [72].

**Figure 9 pharmaceutics-15-02370-f009:**
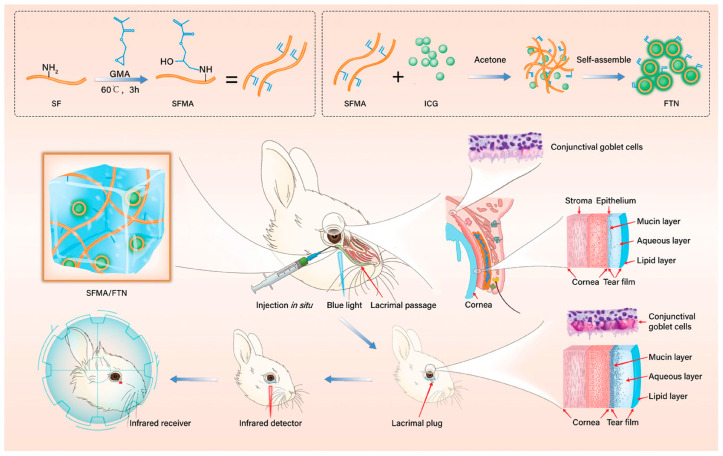
Synthesis and application of SFMA/FTN hydrogel tear plugs. SF, silk fibroin; SFMA, methacrylate-modified silk fibroin; GMA, glycidyl methacrylate; ICG, indocyanine green; FTN, fluorescent tracer nanoparticles. Reproduced with permission from [Dai, M.; Xu, K.; Xiao, D.; Zheng, Y.; Zheng, Q.; Shen, J.; Qian, Y.; Chen, W.], [Advanced Healthcare Materials]; published by [WILEY], [2022] [78].

**Figure 10 pharmaceutics-15-02370-f010:**
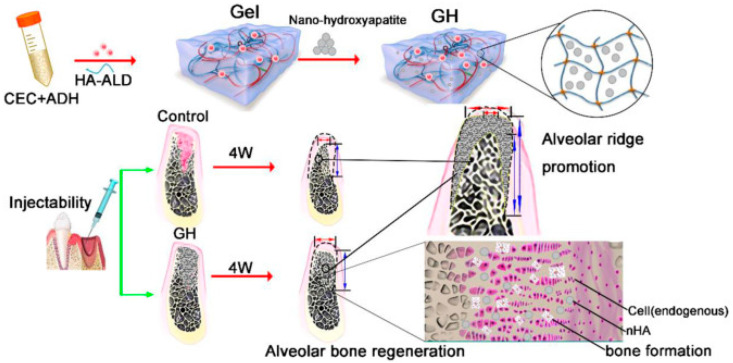
Schematic synthesis of nano-hydroxyapatite hydrogels and their application to the preservation of the alveolar ridge. CEC, N-carboxyethyl chitosan; ADH, adipic acid dihydrazide; HA, hyaluronic acid; ALD, acid-aldehyde; GH, hydrogel-nano-hydroxyapatite. Reproduced with permission from [Pan, Y.S.; Zhao, Y.; Kuang, R.; Liu, H.; Sun, D.; Mao, T.J.; Jiang, K.X.; Yang, X.T.], [Materials Science and Engineering: C]; published by [ELSEVIER], [2020] [91].

**Figure 11 pharmaceutics-15-02370-f011:**
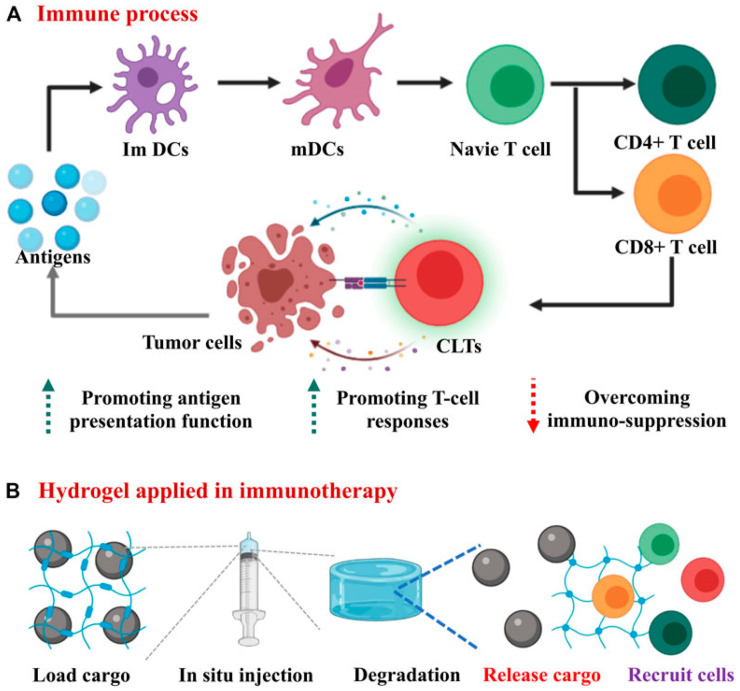
Overview of the application of hydrogels in cancer immunotherapy. (**A**) Schematic of the immune response in vivo; (**B**) hydrogels for immunotherapy. Im DCs, immature dendritic cells; mDCs, mature dendritic cells; CLTs, cytotoxic T lymphocytes. Reproduced from [Cui, R.W.; Wu, Q.; Wang, J.; Zheng, X.M.; Ou, R.Y.; Xu, Y.S.; Qu, S.X.; Li, D.Y.], [FRONTIERS IN BIOENGINEERING AND BIOTECHNOLOGY]; published by [Frontiers], [2021] [111].

**Figure 12 pharmaceutics-15-02370-f012:**
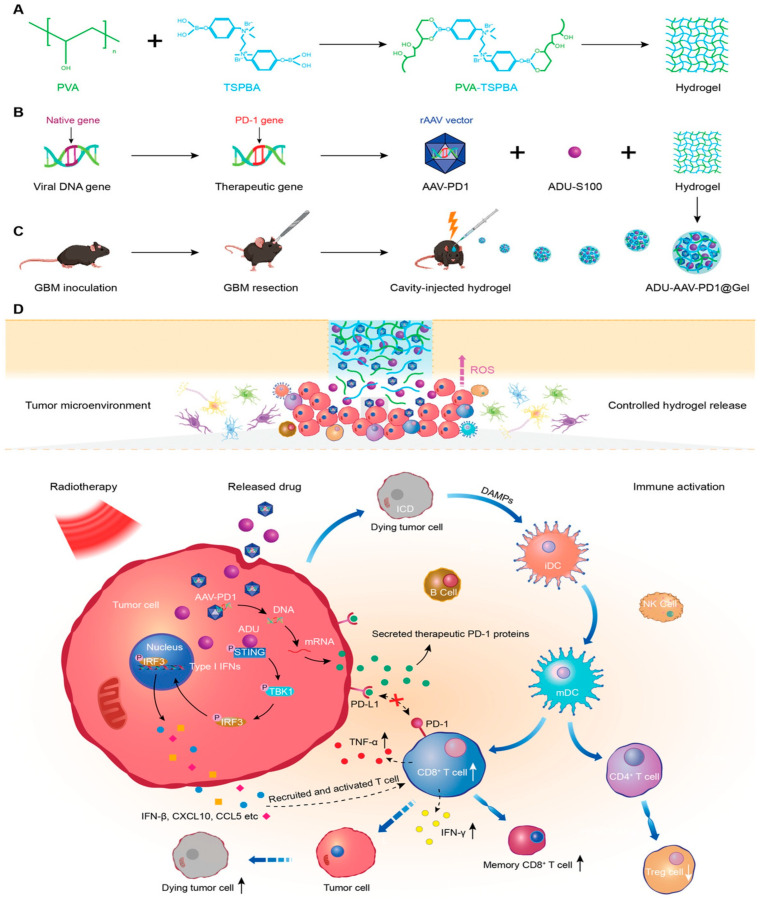
Synthesis and mechanism of ADU-AAV-PD1@Gel. (**A**) Preparation scheme of PVA-TSPBA hydrogel; (**B**) schematic illustration of AAV-PD1 generation and ADU-AAV-PD1@Gel preparation; (**C**) ADU-AAV-PD1@Gel combined with RT in a GBM resection mouse model; (**D**) mechanism of ADU-AAV-PD1@Gel combined with RT in radioimmunotherapy after glioma resection. PVA, poly(vinyl alcohol); TSPBA, N^1^-(4-boronobenzyl)-N^3^-(4-boronophenyl)-N^1^,N^1^,N^3^,N^3^-tetra-methylpropane-1,3-diaminium; AAV, adeno-associated virus; rAAV, recombinant adeno-associated virus; ADU, stimulator of interferon gene (STING) agonist; TBK1, TANK Binding Kinase 1; IRF3, interferon regulatory Factor 3; IFNs, type I interferons; DAMPs, damage-associated molecular patterns. Reproduced with permission from [Sun, S.; Gu, W.; Wu, H.S.; Zhao, Q.Q.; Qian, S.Y.; Xiao, H.; Yang, K.; Liu, J.; Jin, Y.; Hu, C.P.; et al.], [Advanced Functional Materials]; published by [WILEY], [2022] [115].

**Figure 13 pharmaceutics-15-02370-f013:**
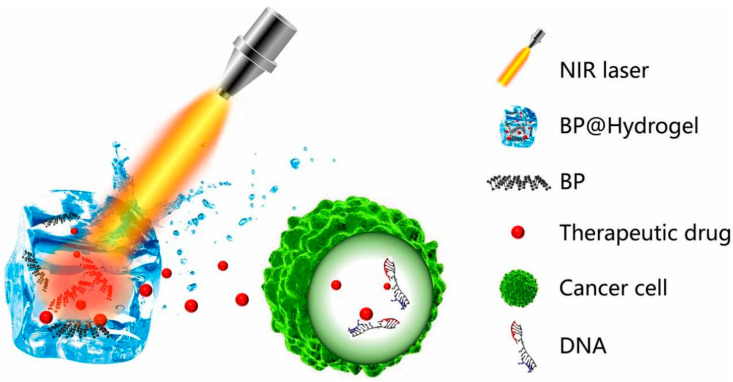
Schematic illustration of the functional principle of the BPNs@Hydrogel. Reproduced from [Qiu, M.; Wang, D.; Liang, W.; Liu, L.; Zhang, Y.; Chen, X.; Sang, D.K.; Xing, C.; Li, Z.; Dong, B.; et al.], [Proceedings of the National Academy of Sciences of the United States of America]; published by [PNAS], [2018] [119].

**Figure 14 pharmaceutics-15-02370-f014:**
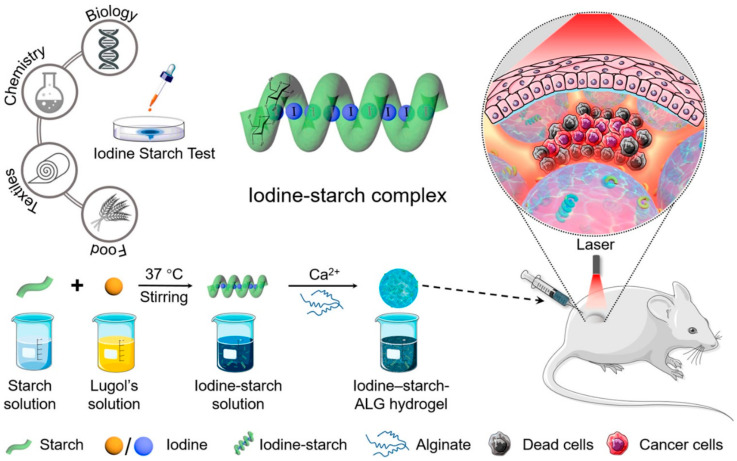
Synthesis of iodine-starch-ALG hydrogels for tumor photothermal therapy. Reprinted with permission from {Wang, H.Y.; Jiang, L.M.; Wu, H.H.; Zheng, W.Y.; Kan, D.; Cheng, R.; Yan, J.J.; Yu, C.; Sun, S.K. Biocompatible Io-dine-Starch-Alginate Hydrogel for Tumor Photothermal Therapy. Acs Biomater. Sci. Eng. 2019, 5, 3654–3662. https://doi.org/10.1021/acsbiomaterials.9b00280}. Copyright {2019} American Chemical Society [124].

**Figure 15 pharmaceutics-15-02370-f015:**
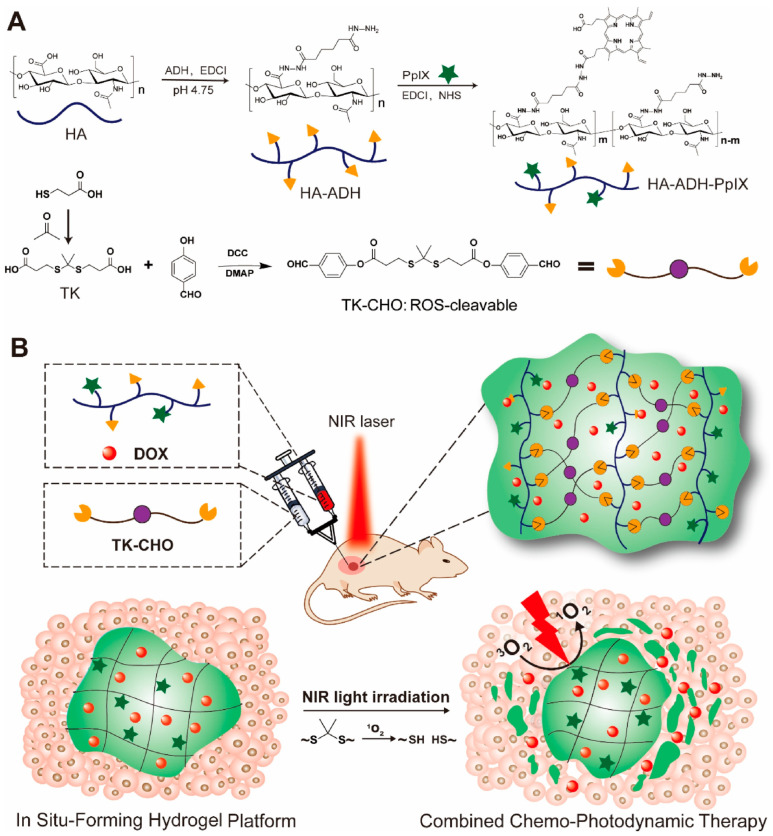
(**A**) Schematic diagram of injectable NIR-triggered ROS-degradable hydrogel synthesis; (**B**) degradable hydrogels for combined chemotherapy and photodynamic therapy. HA, hyaluronic acid; ADH, adipic dihydrazide; EDCI, 1-ethyl-3-(3-(dimethylamino)propyl) carbodiimide hydrochloride; PpIX, photosensitizer protophorphyrin IX; NHS, N-hydroxysuccinimide; TK, thioketal linker; DCC, dicyclohexylcarbodiimide; DMAP, 4-(dimethylamino)pyridine; DOX, doxorubicin. Reproduced with permission from [Xu, X.Y.; Zeng, Z.S.; Huang, Z.Q.; Sun, Y.W.; Huang, Y.J.; Chen, J.; Ye, J.X.; Yang, H.L.; Yang, C.Z.; Zhao, C.S.], [Carbohydrate Polymers]; published by [ELSEVIER], [2020] [134].

**Figure 16 pharmaceutics-15-02370-f016:**
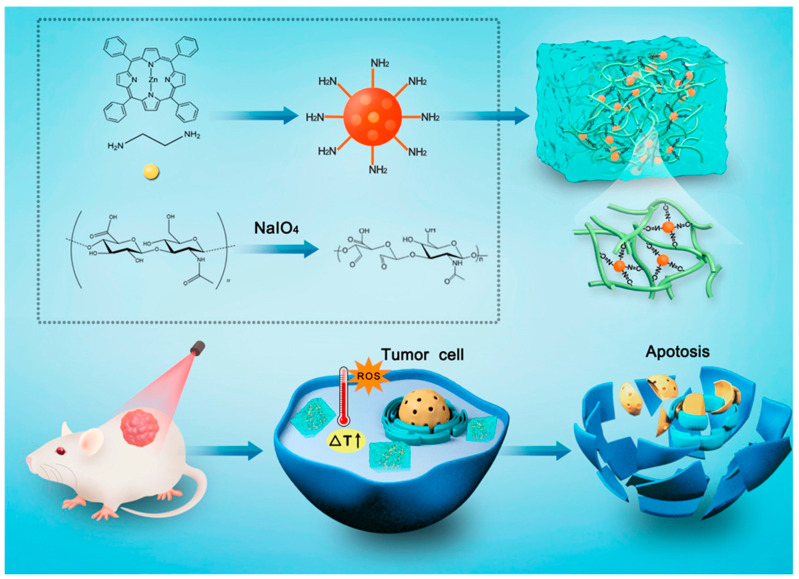
Schematic illustration of the synthesis process of CDs@Hydrogel and its application in cancer therapy. Reprinted with permission from {Yue, J.; Miao, P.; Li, L.; Yan, R.H.; Dong, W.F.; Mei, Q. Injectable Carbon Dots-Based Hydrogel for Combined Photothermal Therapy and Photodynamic Therapy of Cancer. Acs Appl. Mater. Inter 2022, 14, 49582–49591. https://doi.org/10.1021/acsami.2c15428}. Copyright {2022} American Chemical Society [139].

**Figure 17 pharmaceutics-15-02370-f017:**
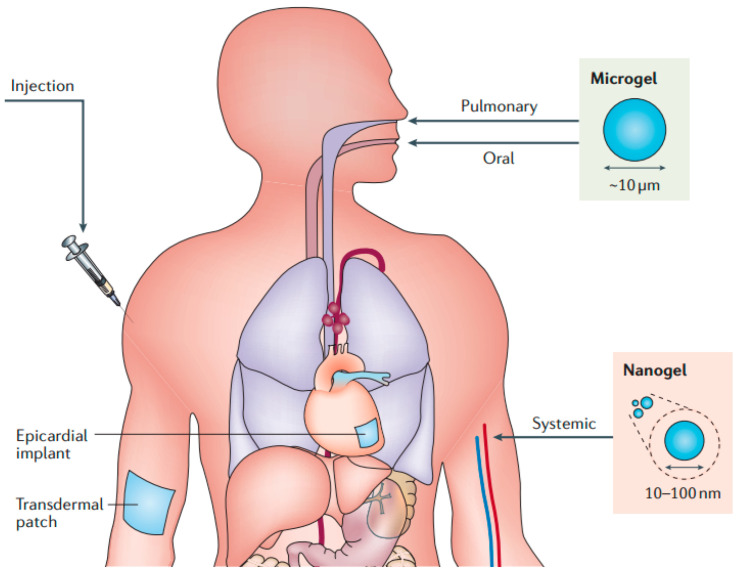
Degradable hydrogel delivery formulations. Reproduced from [Li, J.; Mooney, D.J.], [Nature Reviews Materials]; published by [Nature Publishing Group], [2016] [5].

**Table 1 pharmaceutics-15-02370-t001:** Disease applications and useful characteristics of hydrogels.

Application	Useful Characteristics
Wound Treatment	supports cell adhesion and proliferation, biocompatibility, biodegradability
Tissue engineering	biocompatibility, excellent mechanical properties, biodegradability
Oncology treatment	high water content, drug delivery, biocompatibility, biodegradability
Brain diseases	biodegradability, drug delivery, biocompatibility

## Data Availability

The data presented in this study are available in this article.

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
