# Peer review of "Applications of Degradable Hydrogels in Novel Approaches to Disease Treatment and New Modes of Drug Delivery"

_pharmaceutics, 2023, doi:10.3390/pharmaceutics15102370_

Round 1
Reviewer 1 Report
The manuscript entitled "Applications of degradable hydrogels in novel approaches to disease treatment and new modes of drug delivery", from the authors Bo Hu, Jinyuan Gao, Yu Lu and Yuji Wang.
This review manuscript is very extensive, interesting and comprehensive. The authors have made a complete review of degradable hydrogels on the given topic. Therefore, I consider that this manuscript should be published in the journal "Pharmaceutics" without corrections.
Author Response
Thank you for your recognition.
Reviewer 2 Report
It is an an interesting review.
There are some small problems
In figure 1 at water absorption, water is not completely wrote!
Explain what means: HA, EDC, NHS, HA-DA, PDA and all abbreviations in figure 3.
What means L-ARG? but Zn- MOF? but CPHF? but ADU? Please explain all abbreviations in the article!
Explain in figure captions of figure 7 what means BMSCs.
Explain in figure captions of figure 8 what means U-Arg-PEA.
Explain in figure captions of figure 10 all the abbreviations in the figure. Make the same for Figure captions of Figure 11. Also for figure 12. Also for figure 15.
Please read again the article. There are 2-3 small English problems
Author Response
Dear Editor,
We have resubmitted our manuscript, “Applications of degradable hydrogels in novel approaches to disease treatment and new modes of drug delivery” in Pharmaceutics. We thank you and the reviewer for the constructive comments. The manuscript is revised based on your comments. All changes are highlighted with red font. Here enclosed please also find a point-by-point response to the reviewer's comments.
Please do not hesitate to contact us with any further questions or recommendations.
Yours sincerely,
Yuji Wang
Department of Medicinal Chemistry, School of Pharmacy
Capital Medical University
wangyuji@ccmu.edu.cn
Referee: 2
Comments to the Author
This review manuscript is very extensive, interesting and comprehensive. The authors have made a complete review of degradable hydrogels on the given topic. Therefore, I consider that this manuscript should be published in the journal "Pharmaceutics" without corrections. It is an interesting review. There are some small problems
- In figure 1 at water absorption, water is not completely written!
Response: Thanks for your reminding. We have corrected the spelling.
- Explain what means: HA, EDC, NHS, HA-DA, PDA and all abbreviations in figure 3.
Response: Thanks for your kind comments. We have explained all the abbreviations in Figure 3 and we also added the abbreviations for the whole article.
Abbreviations for figure 3:
|
HA |
Hyaluronic acid |
|
EDC |
1-(3-dimethylaminopropyl)-3-ethylcarbodiimide hydrochloride |
|
NHS |
N-hydroxysuccinimide |
|
DA |
Grafting dopamine |
|
HA-DA |
Hyaluronic acid-graft-dopamine |
|
GO |
Graphene oxide |
|
rGO |
Reduced graphene oxide |
|
PDA |
Polydopamine |
|
HRP |
Horseradish peroxidase |
- What means L-Arg? but Zn- MOF? but CPHF? but ADU? Please explain all abbreviations in the article!
Response: Thanks for your suggestion. We have added the abbreviations in the article.
- Explain in figure captions of figure 7 what means BMSCs.
Response: Thanks for your reminding. BMSCs means bone marrow mesenchymal stem cells. We have explained it in Figure 7.
- Explain in figure captions of figure 8 what means U-Arg-PEA.
Response: Thanks for your reminding. U-Arg-PEA means cationic arginine-based unsaturated poly(ester amide)s. We have explained it in Figure 8.
- Explain in figure captions of figure 10 all the abbreviations in the figure.
Response: Thanks for your reminding. We have explained all the abbreviations. Abbreviations for figure 10:
|
CEC |
N-carboxyethyl chitosan |
|
ADH |
Adipic acid dihydrazide |
|
HA |
Hyaluronic acid |
|
ALD |
Acid-aldehyde |
|
GH |
Hydrogel-nano-hydroxyapatite |
- Explain in figure captions of figure 11 all the abbreviations in the figure.
Response: Thanks for your reminding. We have added all the abbreviations in Figure 11.
Abbreviations for figure 11:
|
Im DCs |
Immature dendritic cells |
|
mDCs |
Mature dendritic cells |
|
CLTs |
Cytotoxic T lymphocytes |
- Explain in figure captions of figure 12 all the abbreviations in the figure.
Response: Thanks for your reminding. We have added it.
Abbreviations for figure 12:
|
PVA |
Poly(vinyl alcohol) |
|
TSPBA |
N1-(4-boronobenzyl)-N3-(4-boronophenyl)-N1,N1,N3,N3-tetra- methylpropane-1,3-diaminium |
|
AAV |
Adeno-associated virus |
|
rAAV |
Recombinant adeno-associated virus |
|
ADU |
Stimulator of interferon gene (STING) agonist |
|
TBK1 |
TANK Binding Kinase 1 |
|
IRF3 |
Interferon regulatory Factor 3 |
|
IFNs |
Type I interferons |
|
DAMPs |
Damage-associated molecular patterns |
- Explain in figure captions of figure 15 all the abbreviations in the figure.
Response: Thanks for your reminding. We have added the abbreviations in Figure 15.
Abbreviations for figure 15:
|
HA |
Hyaluronic acid |
|
|
ADH |
Adipic dihydrazide |
|
|
EDCI |
1-ethyl-3-(3- (dimethylamino)propyl) carbodiimide hydrochloride |
|
|
PpIX |
Photosensitizer protophorphyrin IX |
|
|
NHS |
N-hydroxysuccinimide |
|
|
TK |
Thioketal linker |
|
|
DCC |
Dicyclohexylcarbodiimide |
|
|
DMAP |
4-(dimethylamino)pyridine |
|
|
DOX |
Doxorubicin |
|
- Please read again the article. There are 2-3 small English problems.
Response: Thanks for your kind review. We have corrected the English problems in the article.